# Feasibility Study on the Radiation Dose by Radioactive Magnetic Core-Shell Nanoparticles for Open-Source Brachytherapy

**DOI:** 10.3390/cancers14225497

**Published:** 2022-11-09

**Authors:** Rogier van Oossanen, Jeremy Godart, Jeremy M. C. Brown, Alexandra Maier, Jean-Philippe Pignol, Antonia G. Denkova, Kristina Djanashvili, Gerard C. van Rhoon

**Affiliations:** 1Department of Radiotherapy, Erasmus MC Cancer Institute, University Medical Center, 3015GD Rotterdam, The Netherlands; 2Department of Radiation Science and Technology, TU Delft, 2629JB Delft, The Netherlands; 3Department of Physics and Astronomy, Swinburne University of Technology, Hawthorn 3122, Australia; 4Department of Biotechnology, TU Delft, 2629HZ Delft, The Netherlands

**Keywords:** magnetic nanoparticles, magnetic hyperthermia, brachytherapy, Monte Carlo simulations, thermal ablation, dual-function nanoparticles

## Abstract

**Simple Summary:**

Breast cancer is the most prevalent cancer in women across the world. Most of these patients are diagnosed in an early stage, having only one small tumor. Treatment for these early-stage tumors often includes surgery followed by external beam radiotherapy. While highly effective, this treatment can be time-consuming, taking a few months to complete. This work investigates a new treatment where surgery is replaced by thermal ablation using magnetic nanoparticles. By making these nanoparticles also radioactive, radiotherapy and thermal ablation can be combined in one single treatment. We have investigated the radiation dose profiles of these radioactive magnetic nanoparticles using Monte Carlo computer simulations. It was found that the dose profiles are similar to commercial radioactive sources already used in the clinic. This indicates that the nanoparticles are suited to deliver a clinical dose to the patient, bringing this new treatment a step closer to introduction into the clinic.

**Abstract:**

Background: Treatment of early-stage breast cancer currently includes surgical removal of the tumor and (partial) breast irradiation of the tumor site performed at fractionated dose. Although highly effective, this treatment is exhaustive for both patient and clinic. In this study, the theoretical potential of an alternative treatment combining thermal ablation with low dose rate (LDR) brachytherapy using radioactive magnetic nanoparticles (RMNPs) containing 103-palladium was researched. Methods: The radiation dose characteristics and emission spectra of a single RMNP were calculated, and dose distributions of a commercial brachytherapy seed and an RMNP brachytherapy seed were simulated using Geant4 Monte Carlo toolkit. Results: It was found that the RMNP seeds deliver a therapeutic dose similar to currently used commercial seed, while the dose distribution shows a spherical fall off compared to the more inhomogeneous dose distribution of the commercial seed. Changes in shell thickness only changed the dose profile between 2 × 10^−4^ mm and 3 × 10^−4^ mm radial distance to the RMNP, not effecting long-range dose. Conclusion: The dose distribution of the RMNP seed is comparable with current commercial brachytherapy seeds, while anisotropy of the dose distribution is reduced. Because this reduces the dependency of the dose distribution on the orientation of the seed, their surgical placement is easier. This supports the feasibility of the clinical application of the proposed novel treatment modality.

## 1. Introduction

Breast cancer is the most prevalent cancer in females worldwide, with more than 2.1 million new patients every year [1]. In an effort to combat breast cancer, population screening programs are becoming more common, and the number of patients diagnosed with breast cancer at an early stage is significant. In the USA, 63% of breast cancer diagnosed is still localized, and this patient group has a 5-year survival rate of almost 99% [2]. Traditionally, treatment of early-stage localized breast cancer mostly includes surgery to remove the bulk of the tumor, followed by whole breast external beam radiotherapy to kill any remaining cancer cells. Whole breast irradiation (WBI) is normally given five times a week for 3 to 7 weeks [3]. While this treatment is very effective, it also comes with significant side effects, such as cardiac and skin toxicity [4]. In an effort to reduce toxicity, accelerated partial breast irradiation (APBI) has been gaining popularity in the last two decades. Within APBI, the patient is irradiated for a shorter time compared to WBI and only locally at and around the tumor site instead of the whole breast [5,6]. Currently, APBI is mostly applied by accelerated external beam radiotherapy in which patients typically receive 35 to 40 Gy in 10 or more fractions during 1 or 2 weeks, as described by the meta-analysis of randomized trials comparing WBI vs. APBI in breast cancer by Haussman et al. [7,8].

Brachytherapy has been proposed as an alternative to external beam radiotherapy to perform APBI. Several publications of the GEC-ESTRO members on randomized non-inferiority trials demonstrated that multi-catheter brachytherapy after breast-conserving surgery is an effective option for APBI [9,10]. With brachytherapy the tumor is irradiated from the inside and normal tissue exposure is favorable compared to external beam radiotherapy [11]. As the patient needs to recover from the surgery performed to remove the bulk of the tumor, brachytherapy can only start several weeks after surgery. One option to perform brachytherapy is by applying a high dose rate (HDR), using a highly radioactive seed that is inserted sequentially in different locations of the tumor, delivering a local dose in a few hours [12,13,14]. This short irradiation time is convenient because the dose can be delivered during hospitalization and the treatment is finished in one to a few days.

A second interesting alternative to WBI is the use of low dose rate (LDR) brachytherapy, where many radioactive seeds are implanted in the tumor and deliver a therapeutic dose in weeks to months after implantation [15]. The seeds are placed using hollow needles, typically under ultrasound guidance. This minimally invasive procedure reduces toxicity and can suppress the impact of tumor movement, as the seeds will move with the tumor [16,17]. The risk of seed migration, and especially migration outside of the tumor, which might cause additional dose to healthy surrounding tissue, was shown to be limited [15]. Low-energy photon emitters, especially ^103^Pd (palladium-103) for breast application, are used as a radiation source. The implantation can be performed by a minimally invasive procedure and under local anesthesia. The procedure takes a few hours only. LDR brachytherapy is applied using slowly decaying permanent seed implants, which eventually are left inside the patient as non-radioactive implants after the treatment [15]. Recently, LDR permanent breast seed implants (PBSI) have been demonstrated to offer a more convenient treatment for a selected group of early-stage breast cancer patients, while preserving the therapeutic outcome [16,17].

All the above-mentioned methods require a two-step treatment procedure, i.e., surgical removal of the tumor followed by external beam radiotherapy or brachytherapy after a recovery period from the surgery. New techniques have, however, been proposed to avoid surgery and reduce the time between the treatment of the bulk tumor tissue and radiotherapy. By doing so, the overall treatment time as well as the time the patient spends in the hospital can be shortened. Thermal ablation has been suggested as an alternative to surgical intervention for tumor removal [18]. During thermal ablation the tumor is heated to temperatures from 50 °C and above, where a higher thermal dose gives a larger probability of direct cell kill [19,20]. In general, temperatures above 60 °C are considered to lead to instant cell death [21,22]. Several modalities exist to apply thermal ablation, among which are radiofrequency (RF) heating, high intensity focused ultrasound (HIFU) or magnetic nanoparticle (MNP) heating [23]. Thermal ablation enables minimally invasive tumor eradication without the need of an extensive recovery period for the patient [24]. This creates new possibilities to combine macroscopic tumor removal and local irradiation in one single, integral intervention.

In this study we have investigated the potential of a new minimally invasive treatment of early-stage breast cancer, in which thermal ablation is combined with permanent LDR brachytherapy realized through dual-functional radioactive magnetic nanoparticles (RMNP). The magnetic properties of the RMNP enable the application of MNP heating for thermal ablation resulting in tumor eradication without the need for surgery. The radioactive core of the RMNP acts as a PBSI delivering LDR brachytherapy. This new treatment is anticipated to be performed under local anesthesia in a single-day procedure, and to result in a lower patient burden as well as improved cosmetic outcome because of full breast preservation, while achieving the same clinical outcome.

This novel treatment would not only reduce patient burden but is also expected to significantly reduce the costs of the treatment by reducing OR-time needed and removing the need for external beam radiotherapy. This novel treatment would be beneficial, especially where the availability of external beam irradiation facilities is limited, which causes patients to travel long distances to receive treatment. With combining the whole treatment in a one-day procedure, these patients could be helped tremendously.

This concept can be realized by a combination of iron oxide material with superior magnetic properties and ^103^Pd as a radioactive source based on the clinical evidence of its effectiveness in PBSI [16]. To design such a nanoparticle (NP) for brachytherapy, the dose distribution should be carefully evaluated for suitability for the proposed treatment and compared with current brachytherapy treatment plans. It is expected that the RMNP will slowly diffuse out of the seed after implantation, but the diffusion rate in tissue is currently unknown. This study investigates the static situation in which the RMNPs are located in the seed implant location for the duration of the treatment, i.e., during the first few half-lives of ^103^Pd.

The use of radioactive NPs for open-source brachytherapy has been studied recently both in silico, in vitro and in vivo. Most of the NPs used for brachytherapy are gold-based [25,26], combined with various radioisotopes among which are ^198^Au and ^103^Pd. Recent studies include simulation of dose characteristics of radioactive ^198^Au gold NPs [27,28], focusing on NP cluster size effect and intracellular dose distributions. Gold NPs with a core of ^103^Pd were demonstrated in vivo and showed promising results [29]. It was also demonstrated that closely packed radioactive nanoparticles might enhance radiation dose by up to 20%, but only when inter-NP distance is less than 20 nm [30]. However, these studies focus on injection of NPs, or densely packed NP clusters, as a fluid in the tumor, instead of composing solid gel brachytherapy seeds. This, combined with the lack of data on palladium-iron oxide NPs, shows the need for additional in silico research on the dose distribution of these RMNPs.

In current practice, radioactive brachytherapy sources are placed in a metallic casing. Especially for the low-energy photons, the metallic casing has high attenuation, which can influence the dose distribution. In contrast, the radioactive source in the RMNP is embedded in a very thin iron oxide layer of several nanometers (Figure 1). Therefore, the electrons emitted by ^103^Pd should be considered when determining the RMNP dose characteristics. The differences in dose characteristics between metallic brachytherapy seeds and RMNPs can be assessed by simulations.

## 2. Materials and Methods

To characterize the dosimetric properties of the RMNPs, multiple simulations were performed. An overview of these steps is shown in Figure 2. First, a single NP was simulated to determine the dose distribution on micrometer scale around the NP and assess the separate contribution of both electrons and photons of different energies to the dose. The same simulation setup was used to determine the radiation emitted by the NP after shielding by the iron oxide shell, showing the shielding effect for different particle types and energies. This spectrum was then used for simulating the dose distribution of RMNPs packed in the shape of a commercial brachytherapy seed. Finally, a commercial seed model was also simulated, and the dose distribution was compared with the RMNP seed.

### 2.1. Simulation Setup

The simulations performed in this study consist of two parts. For the first simulation, the Monte Carlo Transport code of Geant4 (version 10.5) [31,32,33] was used. Geant4 is a Monte Carlo simulation toolkit with a wide range of applications from high-energy particle physics to radiation safety; it has been shown to be accurate for medical physics applications [34]. In addition, Geant4 provides the ability to calculate the effects of very thin layers of iron oxide in the NP’s shell, including full atomic de-excitation cascades for ionized atoms [35].

In the second part, after characterizing the radiation dose for a single NP, the macroscopic dose distribution was calculated. Therefore, a brachytherapy ‘seed’ was simulated consisting of a homogeneous mixture of the dual-element NPs. The spectrum of the radiation leaving the NP as calculated in the previous simulation was used as the source spectrum. These simulations were performed using TOPAS MC [36], which serves as an additional layer on top of the Geant4 toolkit and creates a more accessible interface for the user focusing on medical physics specifically. In this research both Geant4 and the TOPAS MC interface were used. The TOPAS interface was used to calculate the dose distribution of the RMNP seed and the commercial seed model.

### 2.2. Single Nanoparticle Simulations

The simulated NP was a sphere with 25 nm radius consisting of a palladium core and shell of iron oxide of various thicknesses. In the simulations, the thickness of the iron oxide layer is decreased from 20 to 0 nm (no iron oxide layer) in steps of 5 nm. For instance, the first geometry has a 5 nm radius palladium core with 20 nm thick iron oxide layer, the second has 10 nm radius palladium core with 15 nm thick iron-oxide layer, etc., and the final geometry has a core of 25 nm radius of palladium without any iron oxide cover. An overview of the NP topologies used in the simulations can be found in Table 1. Both the dose distribution in water and the energy spectrum of the radiation leaving the NPs were assessed. ^103^Pd emits both photons and electrons, mostly with (very) low energy (see Table 2) [37].

Each energy was simulated individually for the single-nanoparticle simulations, in order to be able to compare the effect of the iron-oxide shell for the different electron energies. In the final evaluation, the single energy simulations were combined, after adjusting for the emission intensity of each energy, to provide a spectrum and dose overview of the combined radiation emitted by ^103^Pd.

The simulation parameters were chosen based on the need for nm-scale accuracy and a precise calculation of the auger electrons. The benchmarked G4EmStandardPhysics_option4 physics list was used, as suggested by Arce et al. [34]. The cut length of 1 nm was used for electrons and photons. The low edge energy was set to 10 eV. Full de-excitation was turned on, and Auger and X-ray fluorescence models were used. Multiple scatter was turned off, as it has been shown that the multiple scatter implementation in Geant4 can lead to deviations of more than 20% when simulating low-energy electrons [38]. For the surrounding water the G4_WATER setting was used. The radiation source was configured to irradiate from random locations in the palladium core of the NP. For quantification of the effect of the iron oxide layer on the emitted radiation, simulations were run for the electrons and photons separately. For data processing, the GNU software parallel was used [39].

### 2.3. Brachytherapy Seed Simulations

The simulated RMNP seed consists of a mixture of palladium (5%), iron-oxide (5% iron, 7% Fe_2_O_3_), and water (88%, percentages by mass). The size used for the simulations was corresponding to a commercial seed previously used in the clinic [40], i.e., a 3.14 mm long cylinder with a diameter of 0.826 mm. This seed was placed in a water phantom to calculate the dose distribution and compare it with current brachytherapy seeds, simulated as G4_WATER. The G4EmStandardPhysics_option4 physics list was used, together with the variance reduction method in TOPAS MC to improve statistics further away from the seed. The whole seed was defined as the radiation source, with each particle starting point a random location in the seed. An example of the geometry of the RMNP seed used in the Monte Carlo simulations is shown in Figure 3, which visualizes the seed geometry and a small number of electron and photon absorption paths.

The RMNP seed was designed as a homogenous mixture of radioactive palladium, iron oxide and water. The seed is envisioned to be composed of a solid gel, i.e., agarose, to make the seed sufficiently robust to enable its implantation with conventional brachytherapy equipment. As such a gel is typically composed of water with small amounts of hydrocarbons (<5% mass), the seed was approximated to be water. The surrounding of the seeds was also defined as water. The dose was calculated using two 3-dimensional dose scoring grids. One grid with 0.1 × 0.1 × 0.1 mm voxels served as a high-resolution grid to accurately calculate the dose in the direct surrounding of the seed. The other grid with 0.5 × 0.5 × 0.5 mm voxels calculated the dose in a larger range around the seed.

Finally, a commercially available brachytherapy seed of palladium was simulated using the same software with identical settings to compare the dose distribution with the new RMNP seed. The model used for these simulations was based on the dimensions published by Monroe and Williamson [40]. A schematic drawing of the model, as well as the sizes taken for the seed are shown in Figure 4. For both seeds an (apparent) activity of 70 MBq was taken, which is approximately the activity of palladium LDR brachytherapy seeds used in the clinic. Before calculating the iso-dose lines, a Gaussian filter (σ = 2) was used to filter the noise from the dose distribution.

## 3. Results

### 3.1. Single Nanoparticle Energy Spectra

Figure 5a,b shows the energy spectrum of the electrons and photons when leaving the 5 nm palladium core and 20 nm iron oxide shell NP, respectively. The photon spectrum indicates that the effect of the iron oxide layer of the NP’s energy is negligible: of the more than 100,000 photons included in this simulation, all but a few photons leave the NP with the same energy they had when being emitted from the palladium core. The electron spectrum, however, shows a completely different behavior, as almost all electrons lose part of their energy while passing through the iron oxide layer due to scattering. The result is a broad spectrum of energy of the electrons leaving the NP, with the highest intensity slightly lower than the original energies of the electrons. This effect was observed for all NP core and shell configurations, regardless of the thickness of iron-oxide layer. It can be concluded that the shielding effect of the iron oxide layer on the photon radiation is negligible, while for the electrons it changes the energy spectrum of the radiation significantly.

In Figure 6, the radial radiation dose distribution caused by the electrons, the photons and the total combined dose after 100,000 decays are provided of a NP with a 5 nm palladium core and a 20 nm iron oxide shell. Overall, the electron radiation dose is characterized by a very high and local dose in the immediate surrounding of the NP (30 µm). The small drop in dose at 0.2 µm and around 7 µm corresponds to the range of the lower energy electrons, while at 30 µm, the dose decreases to near-zero as also the higher energy electrons run out of energy. The photon dose shows an exponential decrease with the distance, illustrated by a straight line. At short distance, this dose is insignificant compared to the electron dose, but as soon as the electrons run out of range, the photon dose determines the total dose.

In Figure 7 the dose profiles of three different core-shell topologies are combined in one graph, ranging from 20 nm iron oxide shell thickness to no iron oxide shell. As can be seen from Figure 7a, the dose distribution is almost identical for all three topologies, with a slight difference in the area between 1 × 10^−4^ and 4 × 10^−4^ mm distance. In Figure 7b a zoom-in of this area is shown, where a slight shift in the dose distribution can be observed of about 10^−5^ mm around 2 × 10^−4^ mm radial distance, leading to a local difference in dose of up to 50% maximal. This effect is highly localized, only occurring between 2 × 10^−4^ mm and 3 × 10^−4^ mm radial distance to the NP.

### 3.2. Radioactive Magnetic Nanoparticle (RMNP) Seed

The spectra calculated with the single NP simulations for the 5 nm core 20 nm shell topology were used for the RMNP seed simulations. A cross-section of the total 3D dose distribution of the RMNP seed demonstrates the dose inside the seed as high as 10^5^ Gy, caused mostly by electrons (Figure 8). As the range of the electrons is only a few microns, the dose drops steeply just outside the RMNP seed after which it continues to decay exponentially. From the 2D dose distribution the symmetrical nature of the irradiation can be observed. The lines included in the figure are the iso-dose lines, ranging from 5% dose to 125% dose, where 100% dose is chosen at 2.5 mm from the seed. For the RMNP seed these dose-lines are elliptical, showing an almost symmetrical dose distribution which is slightly elongated in the x-direction close to the RMNP seed, while the iso-dose line is approaching spherical symmetry at larger distances.

The same type of simulation was run with a standard commercial seed model and a 2D cross-section of the dose distribution is shown in Figure 9. The two bright rectangles at the 2D histograms center correspond to the two ^103^Pd plated carbon pellets. The low-dose spot in the middle shows the absorbance of the dose by the lead X-ray marker. The effect of the titanium end cups on the dose distribution can be seen as a dose ‘shadow’ in the shape of a butterfly. The lack of spherical symmetry is clearly visualized by the iso-dose lines, where an underdosage is seen in two diagonal lines, caused by shielding by the titanium encasing. When comparing the iso-dose line of 5%, the RMNP seed delivers 5% dose at a distance of 8 to 9 mm from the seed. For the commercial seed, however, the absorption of the end cups causes a dose fall off to 5% in 6 to 7 mm from the seed’s center, while the 5% dose line extents to more than 10 mm from the seed’s center in the plane perpendicular to the seed’s orientation. This illustrates the less symmetrical behavior of the dose distribution of a commercial seed compared to the RMNP seed, even at larger distances.

## 4. Discussion

The results presented in Figure 8 and Figure 9 show that the RMNP seeds can deliver a therapeutic dose similar to the currently used commercial seeds, while adding a higher symmetry to the dose distribution. These results suggest that an RMNP seed would enable the combination of thermal ablation with LDR brachytherapy in one treatment, without negatively affecting the radiation dose delivered. The shielding effect of the iron oxide layer is negligible on the radiation dose profile, while increasing the efficiency of the seeds and enabling the delivery of the same dose while decreasing the radioactivity needed. In the commercial seed, the spherical symmetry of the radiation dose is limited due to a butterfly effect of the titanium end cups and the absorption of radiation by the lead X-ray marker. The RMNP seed shows a more spherical dose distribution, where the anisotropy is only caused by the cylindrical shape of the seed. Consequently, the orientation of the RMNP is of less importance to the total dose distribution, making their surgical placement easier. The simulation indicates that the RMNP seed can achieve a dose profile similar to the commercial brachytherapy seed. This supports the feasibility of the clinical application of the proposed novel treatment modality.

The simulation data of the dose characteristics of a single NP show a near zero effect of the iron oxide layer on the photon radiation, whereas part of the electrons lose their energy while travelling through the NP. However, this shielding effect on the electron spectra hardly affects the radial dose profile of the RMNP. Only between 200 nm and 300 nm distance of the RMNP the tissue dose profile shows differences between the three shell topologies. As the range of the emitted electrons in water is 30 µm at most, here the electron dose becomes negligible and the X-ray dose dominant. Hence, in the clinical application where the NPs are encapsulated in the RMNP seed, the electrons will not reach outside of the seed and thus will not contribute to the tumor dose.

The simulations also show that within the electron range, the delivered dose is almost exclusively caused by the electrons. In case of diffusion of RMNP outside of the seed, the local dose is expected to be enhanced greatly. If the RMNP stays within the tumor site, the electron energy deposition would help in increasing the local dose without affecting surrounding healthy tissue. However, if the NPs spread outside of the tumor the electron dose would introduce an increased risk of adverse side effects. Predicting the biological effect of the electrons on this scale is complex, as micro-dosimetry can easily be simulated but proves very difficult to be validated experimentally. When applied to patients, the NP properties should be such that migration out of the tumor area is limited within the first few half-lives to limit dose-exposure of healthy tissue.

For this RMNP seed simulation study it is assumed that they do not diffuse out of the seeds. This enables a fair comparison of the radiation characteristics of the NP seeds with the commercial seeds immediately after implantation. However, after some time, it is expected that the NPs will diffuse from the seed matrix into the tumor tissue. Depending on the speed of diffusion, this may lead to different scenarios. When diffusion speed is low (<1 mm/day), the NPs will remain inside the tumor region during the first few half-lives of ^103^Pd, when most of the dose is delivered. Then the slow diffusion of the NPs would spread out the high doses close to the NPs, giving a more homogeneous dose distribution covering the tumor. The higher the diffusion rate, the more spread of the dose will occur. However, when the diffusion rate is increased, the residence time of the NPs in the tumor region decreases. For the treatment to be effective, the diffusion should be slow enough for the NPs to stay in the tumor region during the irradiation, while the spreading of NPs in the tumor tissue should enable a homogeneous dose distribution. Not much literature is available on the diffusion of similar NPs in tissues in general, and breast tissue in particular, which indicates the need for further research into the kinetic behavior of RMNP in tissue.

During the thermal ablation of the tumor, the diffusion speed of the RMNPs might temporarily change as diffusion is temperature dependent. The extent of change depends on the gel used, the temperature reached and the induced structure changes in the tumor microenvironment. The impact of this on the final diffusion is subject to further study.

The RMNP seed is simulated as RMNPs in water, while it is expected that the RMNP seed will be composed of a solid gel, e.g., agarose. Agarose gel and other solid gels are typically made using only a few mass percent of the polymers, with the rest of the gel being water. Moreover, these polymers are made of hydrocarbons, containing only hydrogen, oxygen and carbon, all low-Z elements. Therefore, it is expected that the difference in dose profile of a solid gel seed compared to a water seed is negligible.

This simulation work is based on a spherical RMNP with a palladium core that has an iron oxide shell. However, in practice, the synthesis of NPs could lead to irregular shapes. These were not considered in the current simulation; however, considering the limited effect of the iron oxide layer on the dose distribution it can be assumed that the results of this simulation will remain valid for the irregular NP shapes.

## 5. Conclusions

This study serves as a first step in the development of a new cancer treatment using dual-material radioactive magnetic core-shell NPs. The Geant4/TOPAS Monte Carlo simulation on the radiation dose suggests that a brachytherapy seed composed of NPs containing ^103^Pd core coated with iron oxide shell could give a dose similar to the current commercial brachytherapy seeds. It is demonstrated that the dose distribution of an RMNP seed encapsulated in alginate is determined by the photon component of the ^103^Pd spectrum. Only in cases of individual RMNP, the electron radiation contributes to the dose within a radius of 30 µm of the NP center. The dose distribution of the RMNP seed is comparable with current commercial brachytherapy seeds while the anisotropy of the dose is reduced. Because this reduces the dependency of the dose distribution on the orientation of the seed, their surgical placement is easier. This supports the feasibility of the clinical application of the proposed novel treatment modality.

## Figures and Tables

**Figure 1 cancers-14-05497-f001:**
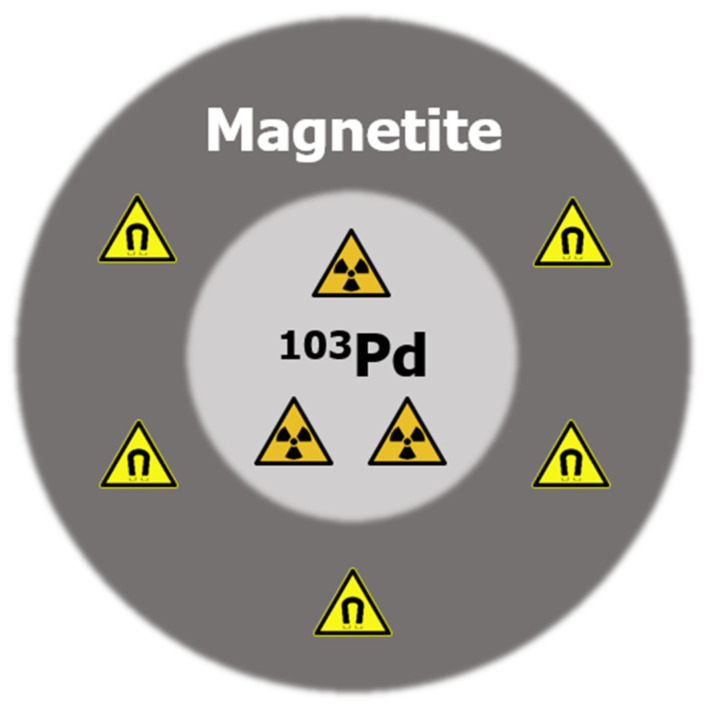
Schematic drawing of the radioactive magnetic nanoparticle, with a core of radioactive ^103^Pd and a shell of magnetic iron oxide.

**Figure 2 cancers-14-05497-f002:**

Overview of the simulations performed to characterize the nanoparticle seeds.

**Figure 3 cancers-14-05497-f003:**
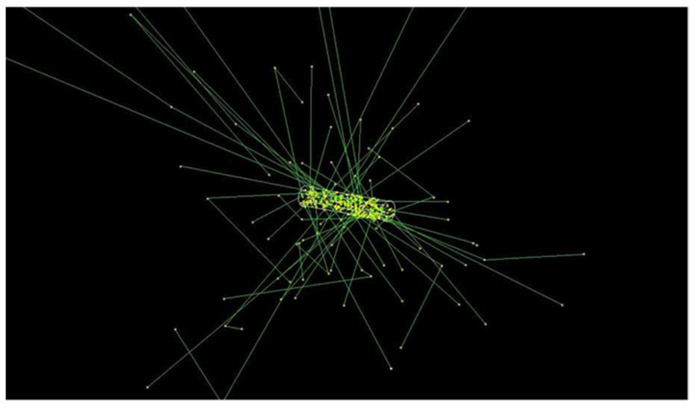
3D visualization of the Monte Carlo simulation of the seed-like structure, showing photons (green) and electrons (red), which are almost invisible due to the short range.

**Figure 4 cancers-14-05497-f004:**
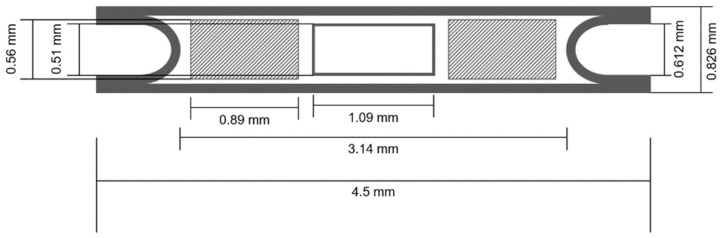
Schematic drawing of the commercial brachytherapy seed indicating the dimension of the different components in millimeters. The diagonally shaded rectangles on the sides represent the ^103^Pd, the white rectangle in the center depicts the lead marker, all encased in a titanium cylinder closed by two arced end cups.

**Figure 5 cancers-14-05497-f005:**
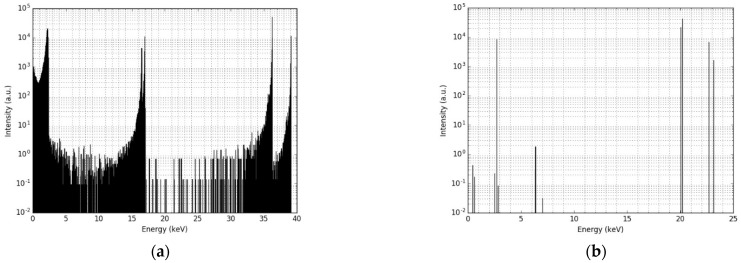
The energy spectra of: (**a**) the electrons leaving a nanoparticle of 25 nm radius (5 nm palladium core plus 20 nm iron oxide shell); and (**b**) the photons leaving the same nanoparticle. The simulation started all particles emitted homogeneously throughout the palladium core in the middle of the nanoparticle.

**Figure 6 cancers-14-05497-f006:**
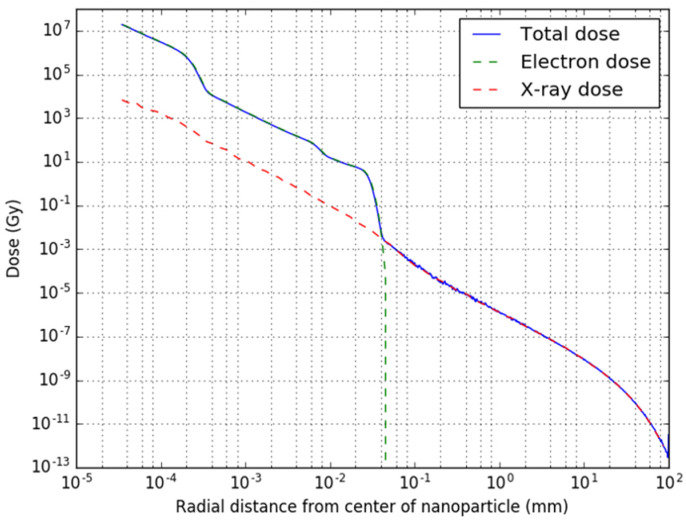
Radial dose profile of the combined radiation of one nanoparticle with 5 nm palladium core and 20 nm iron oxide shell after 100,000 decays.

**Figure 7 cancers-14-05497-f007:**
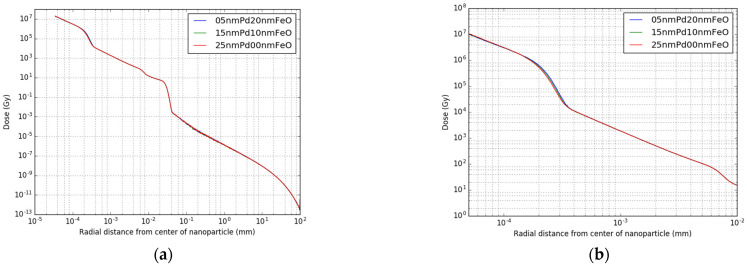
Radial radiation dose profile after 100,000 decays of: (**a**) the combined radiation of one nanoparticle with three core-shell topologies of 20, 10 and 0 iron oxide shell; and (**b**) a zoom-in of the dose profile around 1 × 10^−4^ mm distance.

**Figure 8 cancers-14-05497-f008:**
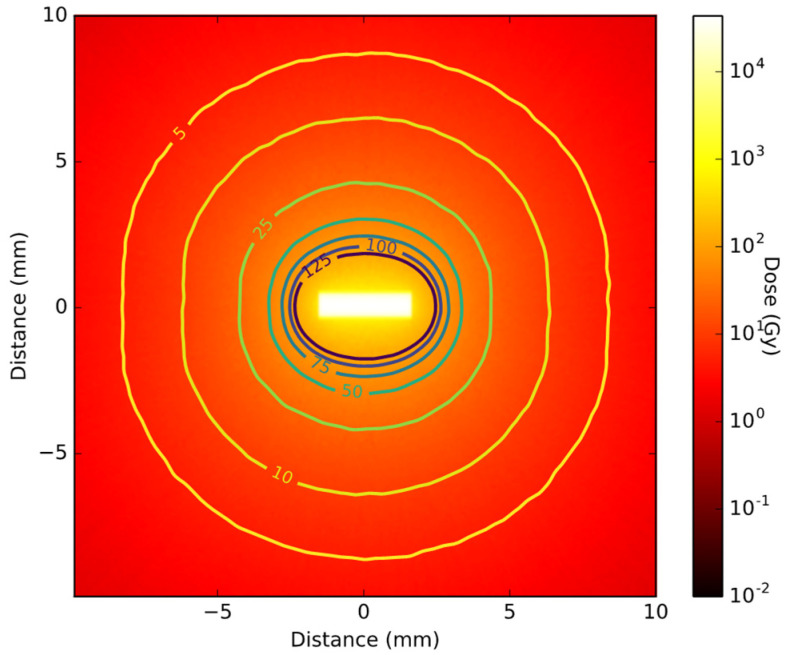
Cross-sectional view of simulated 2D dose distribution of a brachytherapy seed containing nanoparticles with a total ^103^Pd activity of 70 MBq. The iso-dose lines show the dose from 5% to 125%.

**Figure 9 cancers-14-05497-f009:**
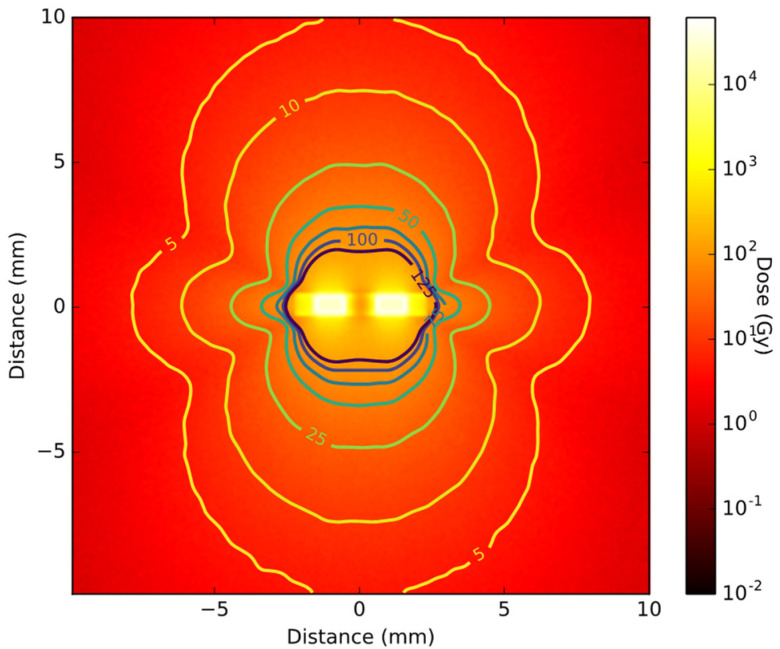
Cross-section view of the 2D dose distribution of the commercial seed, with iso-dose lines ranging from 5% dose to 125% dose.

**Table 1 cancers-14-05497-t001:** Nanoparticle topologies used for the single nanoparticle simulations.

^103^Pd Core Radius (nm)	Iron Oxide Shell Thickness (nm)
5	20
10	15
15	10
20	5
25	0

**Table 2 cancers-14-05497-t002:** Energies of the X-rays and electrons emitted by ^103^Pd and their intensity. Only intensities >1% are included [37].

X-ray Energy (keV)	Intensity (%) ^1^	Electron Energy (keV)	Intensity (%) ^1^
2.7	8.7	2.4	168.0
20.0	22.4	16.5	9.5
20.2	42.5	17.0	18.2
22.7	6.9	36.3	71.2
23.2	1.6	39.1	14.4

^1^ Intensity is given as % emitted per decay, e.g., 200% X-ray emission equals to an average emission of 2 X-rays per decayed atom.

## Data Availability

The data presented in this study are available on request from the corresponding author. The data are not publicly available due to pending patent application.

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
