# Peer review of "Feasibility Study on the Radiation Dose by Radioactive Magnetic Core-Shell Nanoparticles for Open-Source Brachytherapy"

_cancers, 2022, doi:10.3390/cancers14225497_

Round 1

Reviewer 1 Report

This manuscript presents MC based particle scattering simulations with the Geant4/TOPAS framework to evaluate the radiation dose by core-shell nanoparticles for open source brachytherapy.
They authors characterize the energy deposit characteristics of of magnetic core-shell nanoparticles (NP) with a radioactive 103Pd core surrounded by a layer of magnetite.
The thickness of core and layer are varied such that the total diameter stays constant.
In general, the manuscript is concerned with an topic of interest suitable for the readership of cancers.
It is well structured and presents the work in a clear manner.  
However, some major revisions and crucial changes need to be done in the simulations which seem to include some too simplistic models before I can recommend publication of the manuscript.
Furthermore I would suggest some more in detail discussions, explanations and extension regarding the points given below  in the section about the minor issues.

Main issues:

 The most critical point in the paper are related to the simulation of the single core-shell nanoparticles as given in the following:

a) Why did the authors not simulate the radioactive decay of Pd directly, which is possible with the radioactive decay processes in Geant4.

b) Why did the authors only simulate a decay from the center of the NP? The decay position can be expected to influence the results. Random decay positions can be easily simulated within Topas by applying the respective source.
Therefore I recommend to perform the simulations in a more appropriate manner with randomly distributed radioactive decay positions within the radioactive part of the NP.

c) With random decay positions within the radioactive part of the NP the thickness of non-radioactive shell can be expected to influence the energy distribution characteristics, when an appropriately low cutoff length is chosen for the simulations. Therefore the assumptions used in developing the model seem to be somewhat oversimplified to represent an appropriate model for the core shell NP under investigation.

d) Doing this, the thickness of the shell should indeed influence the energy deposit in the vicinity of the NP, which should then be discussed  in detail.

e) Therefore it makes sense to additionally simulate a constant core with increasing shell thickness.

f) Generally, a cut length lower that the shell or core diameter should be used and given in the script.

g) The simulational parameters, materials etc, are given with too little detail to enable reproducibility. Especially the cut length has to be mentioned and the density of the materials.

Some minor issues:

- Table 2  Electron energy intensities are given in % but the values sum up to more than 100%, please correct.

- In sec 2.3 the authors say first that the RNMP seed consists out of palladium airon and water (line 185) and later they refer to iron oxide (line 199), which is correct? Please clarify and correct.

- If I understand it correctly, the mentioned Agarose within the seed wasn't included in the simulation explicitly? Please clarify and discuss the implications in the manuscript.

- Please discuss the important point, how thermal heating to 50C would influence migration of the NP within the implant and possible migration out of it. What are the implications and critical points for the application in vivo, please discuss?

- In Fig 8 and Fig 9, the dose distributions seems not only to be more inhomogenious in the commercial seed compared to the brachytherapy seed, but the isolines seem to vary between the two setups in terms of their average distance. Please discuss this behavior in more detail and try to quantify the differences, since they represent one of the main outcomes of the present study. Compare with other studies.

- Fig 6 and Fig 7, here the dose should given per decay or per x decays, otherwise it can't be compared properly.

- Fig 4 the differently shaded areas are hard to distinguish. If possible, please choose something which is easier to distinguish.

- No other work with respect to previous work with respect to radioactive nanoparticles is given, please provide accurate references and include the respective results in in the introduction and discuss the work:
La Prise Peltier et al Adv. Healthcare Mater. 2018, 7, 1701460 for a general introduction,

- Katti et al Nanotechnol. Biol. Med. 6, 201 (2010),  and other relevant work of Katti et al

- Especially the following literature is of relevance with respect to characterization of radioactive NP and should be discussed:
Hahn et al. Sci Rep 11, 6721 (2021). https://doi.org/10.1038/s41598-021-85964-2 for characterization of radiactive nanoparticles via Geant4, Zutta Villate et al. Eur. Phys. J. D 73, 95 (2019). https://doi.org/10.1140/epjd/e2019-90707-x, Gholami, Y. H., Maschmeyer, R. & Kuncic, Z.  Sci. Rep. 9, 1–13. (2019).

- and applications Chanda, N. et al. Nanomed. Nanotechnol. Biol. Med. 6, 201–209. https://​doi.​org/​10.​1016/j.​nano.​2009.​11.​001 (2010).

Reviewer 2 Report

The article by Rogier van Oossanen et al. “Feasibility Study on the Radiation Dose by Radioactive Mag-netic Core-Shell Nanoparticles for Open-Source Brachytherapy“ presents simulations  using Geant4, a commocially available software package for radiation effect simulations. Approaches for new treatment are calculated where surgery is replaced by brachytherapy using magnetic nanoparticles. These nanoparticles are assumed to be radioactive so that radiotherapy and thermal ablation can be combined. Radiation dose profiles of these radioactive magnetic nanoparticles were investigated using Monte Carlo computer simulations. It was found that the dose profiles are similar to commercial radioactive sources already used in the clinic. The results are a first step towards bringing a new treatment by nanoparticles a step closer to introduction into the clinic.
The article is well written and the results are clearly presented. My major concern refers to the introduction which is very long and redundant. It repeats the abstract, parts of Materials and Methods, and the motivation. This can be shortened considerably.

Author Response

We thank the reviewers for their constructive comments.

In the text hereafter, we provide a point to point response to the comments.

We are grateful for the opportunity to improve the explanation of our research and the results obtained. We hope that we have effectively addressed all concerns of the reviewers and the paper is now acceptable for publication.

Response to reviewer 2:

Comments and Suggestions for Authors

The article by Rogier van Oossanen et al. “Feasibility Study on the Radiation Dose by Radioactive Magnetic Core-Shell Nanoparticles for Open-Source Brachytherapy“ presents simulations  using Geant4, a commercially available software package for radiation effect simulations. Approaches for new treatment are calculated where surgery is replaced by brachytherapy using magnetic nanoparticles. These nanoparticles are assumed to be radioactive so that radiotherapy and thermal ablation can be combined. Radiation dose profiles of these radioactive magnetic nanoparticles were investigated using Monte Carlo computer simulations. It was found that the dose profiles are similar to commercial radioactive sources already used in the clinic. The results are a first step towards bringing a new treatment by nanoparticles a step closer to introduction into the clinic.
The article is well written and the results are clearly presented. My major concern refers to the introduction which is very long and redundant. It repeats the abstract, parts of Materials and Methods, and the motivation. This can be shortened considerably.

Dear reviewer, thank you for the comments.

We have removed a redundant paragraph in the introduction, and also removed some redundant information in the materials and methods part. Because of comments from other reviewers however, several other paragraphs were added to the introduction. We hope the introduction in this new version gives a better overview of the literature related to our research and explains in a clear way our novel approach of the dual-material nanoparticles.

Reviewer 3 Report

The manuscript by van Oossanen is an interesting investigation (simulation) about the thermal ablation using magnetic nanoparticles for breast cancer treatment. The topic of the paper is of interest for scientists working in a multi-disciplinary field and well fits with the scope of Cancers. The experimental design is adequate for achieving the results, which are found to agree with the authors’ hypothesis.

This reviewer positively evaluates the paper, which should be eventually published, but is suggesting some minor revisions before further processing. In details, authors are encouraged to revise the abstract by inserting some key results data. Number can give a direct indication about the importance of the paper in the field. Moreover, in the introduction, although the study aim and the state of the art are well presented, the presentation of the novelty should be improved to strengthen the interest for readers.

Author Response

We thank the reviewers for their constructive comments.

In the text hereafter, we provide a point to point response to the comments.

We are grateful for the opportunity to improve the explanation of our research and the results obtained. We hope that we have effectively addressed all concerns of the reviewers and the paper is now acceptable for publication.

Response to reviewer 3:

The manuscript by van Oossanen is an interesting investigation (simulation) about the thermal ablation using magnetic nanoparticles for breast cancer treatment. The topic of the paper is of interest for scientists working in a multi-disciplinary field and well fits with the scope of Cancers. The experimental design is adequate for achieving the results, which are found to agree with the authors’ hypothesis.

This reviewer positively evaluates the paper, which should be eventually published, but is suggesting some minor revisions before further processing. In details, authors are encouraged to revise the abstract by inserting some key results data. Number can give a direct indication about the importance of the paper in the field. Moreover, in the introduction, although the study aim and the state of the art are well presented, the presentation of the novelty should be improved to strengthen the interest for readers.

Dear reviewer, thanks for the comments.

As recommended we have added a sentence in the abstract reflecting our results on the single nanoparticle simulations, namely:

Changes in shell thickness only changed the dose profile between 2×10-4 mm and 3×10-4 mm radial distance to the NP, not effecting long-range dose. (line 33-34)

To put more emphasize on the novelty of the treatment, we have added the following paragraph to the introduction:

This novel treatment would not only reduce patient burden, but is also expected to significantly reduce the costs of the treatment by reducing OR-time needed and removing the need of external beam radiotherapy. This novel treatment would be beneficial especially where the availability of external beam irradiation facilities is limited, causing patients to travel long distances to receive treatment. With combining the whole treatment in a one-day procedure, these patients could be helped tremendously. (lines 111-116)

Round 2

Reviewer 1 Report

The authors extended the manuscript and addressed most of the concerns previously raised.

I still believe that the manuscript would benefit from some additional discussion which I commented on in the first report.

This is especially the case for question d) about the effects of the effects of the shell thickness.

However this should be seen merely as a recommendation and not as an critical issue.